# Promising Cytomegalovirus-Based Vaccine Vector Induces Robust CD8^+^ T-Cell Response

**DOI:** 10.3390/ijms20184457

**Published:** 2019-09-10

**Authors:** Jian Liu, Dabbu Kumar Jaijyan, Qiyi Tang, Hua Zhu

**Affiliations:** 1School of Biological Sciences and Biotechnology, Minnan Normal University, Zhangzhou 363000, China; 2Department of Microbiology, Biochemistry and Molecular Genetics, Rutgers—New Jersey Medical School, Newark, NJ 07103, USA; 3Department of Microbiology, Howard University College of Medicine, Washington, DC 20059, USA; 4College of Life Sciences, Jinan University, Guangzhou 510632, China

**Keywords:** cytomegalovirus, CMV, HCMV, vaccine vector, T-cell response, vaccine strategy, animal model, disease control

## Abstract

Vaccination has had great success in combating diseases, especially infectious diseases. However, traditional vaccination strategies are ineffective for several life-threatening diseases, including acquired immunodeficiency syndrome (AIDS), tuberculosis, malaria, and cancer. Viral vaccine vectors represent a promising strategy because they can efficiently deliver foreign genes and enhance antigen presentation in vivo. However, several limitations, including pre-existing immunity and packaging capacity, block the application of viral vectors. Cytomegalovirus (CMV) has been demonstrated as a new type of viral vector with additional advantages. CMV could systematically elicit and maintain high frequencies of effector memory T cells through the “memory inflation” mechanism. Studies have shown that CMV can be genetically modified to induce distinct patterns of CD8^+^ T-cell responses, while some unconventional CD8^+^ T-cell responses are rarely induced through conventional vaccine strategies. CMV has been used as a vaccine vector to deliver many disease-specific antigens, and the efficacy of these vaccines was tested in different animal models. Promising results demonstrated that the robust and unconventional T-cell responses elicited by the CMV-based vaccine vector are essential to control these diseases. These accumulated data and evidence strongly suggest that a CMV-based vaccine vector represents a promising approach to develop novel prophylactic and therapeutic vaccines against some epidemic pathogens and tumors.

## 1. Introduction

Cytomegalovirus (CMV) belongs to the virus family *Herpesviridae* and is a ubiquitous β-herpesvirus with a global seroprevalence of more than 50% [1,2]. CMV is an opportunistic pathogen that can establish life-long latency in infected people after primary infection and occasionally reactivate. Both primary infection and reactivation are asymptomatic in healthy individuals, but CMV infection is known to cause congenital infection and other life-threatening consequences in immunosuppressed patients [3,4]. Development of a prophylactic vaccine against CMV infection is a major public health priority [5]. During the past several decades, great efforts have been made to develop an effective CMV vaccine. Various forms of candidate CMV vaccines, including recombinant subunit vaccines, enveloped virus-like particle (eVLP) vaccines, vectored CMV vaccines, nucleic-acid-based CMV vaccines, whole-virus-based attenuated vaccines, and peptide vaccines are under intensive pre-clinical or clinical research [6]. Despite the fact that no CMV vaccine has yet been licensed, great progress has been achieved to better understand the virus.

Owing to the accumulated knowledge toward CMV and the remarkable development of CMV genome engineering, CMV virus is easy to be attenuated and re-designed to improve the safety through reverse genetic manipulation. In recent years, using CMV as a vaccine vector has attracted much attention. Comparing with other viral vectors, CMV has some unique properties that make it be an attractive vaccine vector [7]. The genome of the CMV virus is large (~230 KB) and contains many non-essential genes; therefore, multiple foreign genes can be inserted into the CMV genome without affecting virus replication [8]. Moreover, some well-characterized CMV genomes have been cloned as bacterial artificial chromosomes, making mutagenesis of the viral genome and construction of recombinant viruses easier [8,9,10]. Primary CMV infection and virus reactivation in healthy individuals are asymptomatic, and occasional reactivation continues boosting host immunity through a process known as “memory inflation” and maintaining high frequencies of effector T cells in circulation [11]. CMV is able to achieve secondary infection or superinfection regardless of the status of prior CMV immunity [12].The current review focuses on the progress that has been made toward whole-virus-based CMV vaccines and CMV-based vaccine vectors, which might represent a novel approach to develop vaccines against some pathogens, cancers, and other diseases.

## 2. Whole-Virus-Based CMV Vaccines

Life-long disability associated with congenital CMV infection is highly costly to society and affects social development of these children. The affordable protection for neonates against congenital CMV infection is a major goal of CMV vaccines. CMV is a complex virus with a large genome (~230 KB), expressing hundreds of proteins [8]. The immunogenic proteins of CMV that elicit protective immune responses have not been well-characterized. The whole-virus-based vaccine contains a full set of viral proteins and is more likely to induce immune responses resembling those of natural infection. Live-attenuated vaccines are the most common vaccines and they are effective against many pathogen infections. Human CMV (HCMV) laboratory strains, Towne, and AD169 are the best-characterized strains, derived from extensive passages in fibroblast cells [13,14]. Both viruses were safe and well-tolerated when tested in clinical trials [14,15]. However, these vaccines failed to protect renal-transplant patients and seronegative women against primary infection or viral reactivation [16,17].

Suboptimal efficacy of the live-attenuated vaccines using laboratory strains Towne and AD169 demonstrates the insufficient immunogenicity of the over-attenuated viruses achieved through the fibroblast-adapted process. To generate new live-attenuated-vaccine strains retaining excellent safety and an improved immunogenicity profile, the Towne/Toledo chimeric viruses were constructed by replacing parts of the Towne strain genomic segments with segments from the non-attenuated Toledo strain. These chimeric viruses were evaluated in both seropositive and seronegative individuals in separate Phase I trials. In both trials, the chimeric-vaccine candidates were proved to be well-tolerated and did not cause serious local or systematic infections [18,19]. Out of the enrolled seronegative men, 30.6% (11/36) underwent seroconversion, and most (10/11) occurred in men inoculated with Chimeras 2 and 4 [18]. All seroconverts developed detectable levels of neutralizing antibodies, and some (7/11) developed CD8^+^ T-cell responses to HCMV IE1 protein [18]. Due to the attenuation mechanism not having been fully elucidated, extensive clinical trials are required to demonstrate the safety profile and immunogenicity of these attenuated and non-attenuated chimeric viruses.

Compared to the genome of the fibroblast-adapted virus with the parental clinical strain, several potential attenuating mutations have been characterized [20]. A ~15 KB segment located at the UL/*b*’ region was found to be missing in the genome of fibroblast-adapted strains. This missing segment was demonstrated to be a major attenuation contributor [10]. Deletion of the segment from the pathogenic Toledo strain deprived it of its ability to replicate in human tissue implanted in SCID mice [10].

Another mutation has attracted extensive interest in recent years. It is regarded as a significant contributor to attenuation [21]. Both the fibroblast-adapted Towne strain and the fibroblast-adapted AD169 strain were found to have some mutations in the components of the gH/gL/UL128–131 pentameric complex [22]. Although mutations occur at different genes, these mutations disrupt the formation of the pentamer. The mechanisms of the mutation-prone pentamer when passaged in the fibroblast cells are still not well-elucidated. The pentamer was found to be a major determinant for cell tropism, and the fibroblast-adapted virus with the disrupted pentamer was found to lose its ability to infect a wide range of cell types, including epithelial and endothelial cells, monocytes, and placenta cytotrophoblasts [23]. Moreover, the pentamer is a protein complex with high immunogenicity. When comparing the neutralizing antibody pool elicited by natural infection and immunization of the fibroblast-adapted virus, the two virus types elicited comparable neutralizing-antibody titers against the infection of fibroblast cells; however, the titer of the neutralizing antibodies elicited by natural infection to inhibit the viral infection of epithelial cells was around 28-fold higher than antibodies elicited by the fibroblast-adapted virus [24]. The pentameric complex presents dominant native neutralizing epitopes and antibodies specific to the pentamer that can inhibit viral entry to epithelial cells [25]. Neutralizing antibodies blocking the infection of epithelial and endothelial cells are important for the prevention of primary and intrauterine infection. Maternal antibodies specific to the pentamer were proved to be able to significantly reduce the risk of fetal HCMV transmission during primary infection [26]. CMV with a restored pentamer recovered epithelial tropism and showed improved immunogenicity in rabbits and rhesus macaques [27].

This strategy was adopted to construct a novel whole-virus-based CMV vaccine by Merck & Co. In this form of the candidate vaccine, the mutation causing a frameshift at *UL131* of the fibroblast-adapted AD169 strain was repaired, and the pentameric complex was restored to enhance immunogenicity. Moreover, the destabilizing domain of FK506-binding protein 12 (ddFKBP) was fused to essential viral proteins, pIE1/2 and pUL51 to render the virus conditionally replication-defective [23]. DdFKBP is a destabilizing protein domain; when it is fused to the target protein, the fusion protein will be rapidly degraded by proteasome after it is synthesized; Shield-1 is a small synthetic ligand that can specifically bind to ddFKBP and stabilize the fusion protein [28]; *IE1* and *IE2* are immediate-early genes, and the pIE1 and pIE2 are involved in gene regulation through interaction with basal transcriptional machinery and cellular transcription factors [29,30,31]; and *UL51* is a late gene that is essential for viral genome cleavage and packaging [32]. Both the ddFKBP-IE1/2 and the ddFKBP-UL51 fusion protein function normally in the presence of Shield-1, and the replication of the CMV vaccine strain is not disturbed when propagated in vitro with Shield-1. Since Shield-1 does not exist in vivo, when inoculated with the vaccine virus, ddFKBP-IE1/2 and ddFKBP-UL51 are degraded in the absence of Shield-1; thus, the replication of the virus is disrupted and no progeny virus is generated. Progeny viruses produced in the presence of Shield-1 retain the ability to attach themselves onto and enter susceptible cells even in the absence of Shield-1 during a single-cycle infection, and viral proteins are able to elicit a full repertoire of antibodies and a broad range of T-cell responses specific to different viral proteins [32]. This candidate vaccine is currently evaluated in the Phase II clinical trial. Disclosed data from the Phase I clinical trial showed the vaccine was well-tolerated and it could elicit comparable neutralizing antibodies levels and cellular immune responses with natural CMV infection [33].

## 3. CMV-Vectored Vaccines against HIV

Global-health concerns related to human-immunodeficiency-virus (HIV) infection and acquired immunodeficiency syndrome (AIDS) are still not addressed. Although combination antiretroviral therapy (ART) can lower the virus load, limit transmission, and greatly prolong the life of patient, however, only 59% of patients had received ART globally by the end of 2017 [34]. The HIV epidemic is still ongoing due to infected individuals who are unaware of the infection and/or who not receiving antiviral therapy. Development of effective prophylactic vaccines is very important for enhancing herd immunity and reducing HIV transmission rates.

It is very difficult to control these viruses through the regular immunological mechanism due to the virus being able to evade host immunity and destroy immune cells. During the past four decades, investigators have attempted to stimulate HIV-specific immunity with multiple vaccine strategies [35,36]. At least three vaccine strategies have been tested for vaccine efficacy in clinical trials. One strategy focused on eliciting neutralizing antibodies with HIV envelope glycoproteins (Env) to block infection; another strategy focused on inducing T-cell responses with conserved, internal viral proteins (such as HIV gag, nef, and pol); the third strategy tried to generate both humoral immunity and cellular immunity through heterologous prime/boost procedure (such as prime with a DNA vaccine and boost with an Ad vaccine vector) [37,38,39,40]. However, efficacy testing of all these strategies failed in the clinical trials. These vaccine forms could stimulate humoral immunity and/or cellular immunity to some extent, and some even showed promising immunogenicity in early-phase trials, but all failed to clear the infection [39]. One explanation for these failures is that the elicited immune responses were too late to eliminate virus replication, and the viruses could effectively evade host immunity once infection was established due to their hypervariation and other incompletely known mechanisms [41,42]. Elimination of viruses at the early-replication phase has been demonstrated to be much more effective [43]. An alternative vaccination strategy eliciting unconventional immune responses, which viruses have not learned how to evade, might be a rational effective approach.

CMV is an attractive vaccine vector since its infection could elicit broad and robust immune responses involving both humoral and cellular immunity [11,23]. CMV as a vaccine vector has been demonstrated to be a promising approach in HIV vaccine development. In a study, fibroblast-adapted laboratory Rhesus cytomegalovirus strain (RhCMV68-1) was used as a vector to express simian immunodeficiency virus (SIV) proteins. This vaccine form could provide the early control of infection and persistent protection for 55% rhesus macaque when challenged with highly pathogenic SIVmac239 virus [44]. Further analysis found lymphatic and hematogenous viral dissemination could be detected after SIVmac239 virus challenge, but rhesus macaques that were protected lost signs of infection over time, and SIV RNA or DNA sequences could not be detected above background levels for as long as 69–172 weeks post-challenge using ultrasensitive assays. These results suggested the immune responses elicited by RhCMV/SIV could protect rhesus against highly pathogenic SIV infection in spite of viral dissemination was detected after SIV challenge [45]. This protection represents a promising pathway to use CMV as a vaccine vector for developing both prophylactic and therapeutic vaccines against HIV.

The types of protective immunity induced by RhCMV68-1/SIV were further analyzed to unravel the mechanism of protection. In general, humoral and cellular immunity are two major protective immunity types, commonly elicited after natural infection and vaccine immunization. Components of humoral immunity, especially neutralizing antibodies, usually bind to the membrane proteins of virions and block the entry of viruses. Cellular immunity responds to cells displaying epitopes in the context of MHC molecules. Cellular immunity could directly destroy infected cells by CD8^+^ T cells and play roles in immunomodulation through releasing cytokines by CD4^+^ T cells. When immunizing with RhCMV68-1/SIV, neutralizing antibodies were not detected and were excluded from immune mediators of protection [46]. A persistent SIV-specific CD8^+^ T-cell response was demonstrated to play an essential role in protection; the T-cell response type was induced in all rhesus vaccinated with RhCMV68-1/SIV [44,46]. Further analysis showed that CD8^+^ T-cell responses elicited by RhCMV/SIV were distinct in both quality and quantity from the responses elicited by SIV or other conventional Adenovirus (Ad) or Modified Vaccinia Ankara (MVA) vaccine vectors. In general, the majority of adaptive CD8^+^ T-cell responses elicited by natural HIV infection or conventional vaccines are canonical CD8^+^ T-cell responses restricted by polymorphic MHC-Ia, while CD8^+^ T cells recognizing peptides in the context of MHC-II or highly conserved MHC-E are elicited in rare cases [47,48]. Strikingly, in the rhesus vaccinated with the RhCMV68-1/SIV vector, approximate 70% of the SIV-specific CD8^+^ T-cell responses were unconventional and restricted by MHC-II, while the remaining 30% were conventional and restricted by MHC-I. Moreover, CD8^+^ T-cell responses elicited by RhCMV68-1/gag could not be boosted by SIV infection or a conventional viral vector (Ad or MVA) expressing the same antigen, indicating that the conventional CD8^+^ T-cell response induced by RhCMV68-1/SIV was non-canonical and targeted entirely different epitopes [49]. Furthermore, the epitope-targeting profiles of SIV gag-specific CD8^+^ T cells were analyzed, and results showed that the peripheral blood CD8^+^ T cells elicited by RhCMV68-1/gag recognized as many as three times epitopes (minimum of ~32 distinct epitopes) than those of natural infection or conventional vaccine vaccination (~9–14 distinct epitopes) [49].

The unique characteristic of RhCMV68-1 vectors to elicit MHC-II-restricted CD8^+^ T-cell responses and non-canonical CD8^+^ T-cell responses targeting a different set of epitopes suggests that CMV could modulate and redirect the host recognition mode. Rh182, 184, 185, and 189 proteins (HCMV *US6* gene-family homologs) could cooperate to degrade the heavy chain of nascent MHC-I and prevent the loading of peptides to polymorphic MHC-Ia in the endoplasmic reticulum (ER) [50,51,52], which could prevent the recognition and priming of the canonical CD8^+^ T cell. Moreover, results from assessment of sub-region deletant vectors (△*Rh182-185* and △*Rh186-189*) showed some other mechanisms also play important roles in modulating the canonical CD8^+^ T cells response. Since only deletion of the *Rh189* (HCMV US11 homolog, belongs to *US6* gene family) from RhCMV vector was able to induce canonical MHC-I-restricted CD8^+^ T cells response, despite *Rh182–185* showing higher ability to downregulate MHC-I than *Rh189* [49]. Conventional MHC-I-restricted non-canonical CD8^+^ T cell-responses are rarely induced by other viral-vector platforms due to the special role of the CMV *US6* gene-family homologs.

The gH/gL/UL128–131 complex is essential for both HCMV and RhCMV to infect endothelial, epithelial, and myeloid cells, while this pentamer is not essential for the growth of both HCMV and RhCMV in fibroblast cells. *UL128–131* genes will be spontaneously mutated by serial passages of wildtype CMV virus in fibroblast cells. The HCMV gH/gL/UL128–131 complex has been intensively investigated and proven to be very important for HCMV vaccine efficacy. In the case of HCMV, multiple research papers found the fibroblast-adapted AD169 strain with the pentamer restored could elicit robust neutralizing antibodies compared with natural infection, which could inhibit the viral infection of both fibroblast and epithelial cells [23,25,27,53]. Moreover, the pentamer was also found to play important roles in regulating T-cell immunity in the case of RhCMV. The fibroblast-adapted RhCMV68-1 strain, with spontaneously mutated *UL128* and *UL130* homologs, could elicit broad unconventional CD8^+^ T-cell responses, and these rare T-cell responses are believed to be induced by peptides in the context of MHC-E or MHC-II molecules [49,54]. Moreover, RhCMV68-1 with additional US11 homolog deletion induced both a conventional canonical and an unconventional CD8^+^ T-cell response, and it could not elicit a conventional non-canonical CD8^+^ T-cell response [49,55]. In contrast, the RhCMV strain with restored *UL128* and *UL130* could only elicit conventional MHC-I restricted canonical CD8^+^ T-cell responses, a situation quite similar to wildtype RhCMV [49]. These findings demonstrated at least four different CD8^+^ T-cell responses that could be elicited by CMV vaccine vectors [55]. CMV-based vector is a unique viral vector and it can be genetically manipulated to achieve the stimulation of distinct patterns of T-cell response.

## 4. CMV-Vectored Tuberculosis (TB) Vaccine

Despite a prophylactic vaccine (bacilli Calmette-Guerin, BCG) and a large number of anti-TB drugs that are available to prevent or treat TB, it still remains a leading global cause of morbidity and mortality. The situation is caused by complex reasons. Mycobacterium TB (Mtb) has coevolved with humans for around 70,000 years, and it has evolved many immune evasion strategies to achieve an optimal balance between pathogenicity and host immunity [56,57]. BCG is the only licensed vaccine for TB, and it provides protection against tuberculous meningitis, severe military TB, and some other childhood infections [58]. However, the vaccine efficacy of BCG varies geographically, and BCG is also suboptimal to reduce the incidence of pulmonary TB in adults [59]. Moreover, the speed of emergence of drug-resistant Mtb strains is faster than that of new-drug development. According to the World Health Organization (WHO) Global TB Report 2018, mycobacteria have established a latent infection in around one-fourth of the world population, and TB affects as many as one-third of the world population. New effective vaccines and drugs are a high healthcare priority to combat TB spread.

Using CMV as a vaccine vector to deliver TB antigens is a new approach to prevent Mtb infection. As previously reported, the vaccination of RhCMV/SIV could elicit conventional “non-canonical” MHC-I-restricted and unconventional MHC-II- and MHC-E-restricted CD8^+^ T-cell responses, recognizing unusual, diverse, and highly promiscuous epitopes [45,49]. These T-cell responses were believed to provide promising protection against highly pathogenic SIVmac239 virus infection. Same with HIV, Mtb also developed many immune-evasion strategies during its long-term co-evolution with humans. Whether the CMV-based vector can elicit an unconventional immune response against Mtb infection is a new interest to investigators. A modified (m1–m16-deleted) murine CMV (MCMV) expressing Mtb antigen 85A (MCMV85A) was evaluated in the mouse model, and results showed that immunization with MCMV85A could significantly reduce the mycobacterial load after challenge with Mtb [60]. Analysis showed that MCMV85A was able to elicit a low frequency of 85A-specific T-cell responses in mice, but the expression of IL-21 was also enhanced. Early inhibition of Mtb growth in the lungs of MCMV85A-immunized mice was abolished by NK-cell depletion [60]. These results showed that the innate immune response was playing a much more important role than the Mtb antigen-specific T-cell response in protection against Mtb infection in mice model.

In the rhesus macaque primate model, RhCMV was used as a vector to express Mtb antigens (RhCMV/TB). Highly effector-differentiated, circulating, and tissue-resident Mtb-specific CD4^+^ and CD8^+^ memory T-cell responses were induced in rhesus macaques vaccinated with RhCMV/TB. RhCMV/TB-vaccinated and -unvaccinated rhesus macaques were challenged with highly pathogenic Mtb Erdman strain one year post-vaccination, and the overall extent of Mtb infection and disease was significantly reduced by 68% compared with the unvaccinated group. Approximately 41% of vaccinated rhesus macaques were completely protected from the development of TB [61]. Both Mtb-specific memory T-cell responses and the vaccine-related conductive innate immune state contributed to the protection. Although it remains to be further verified, neutrophils were demonstrated by some reports to be able to kill and restrict the growth of Mtb. In the study of a rhesus macaque vaccinated with RhCMV/TB, neutrophil gene-expression patterns were found to be altered; primed neutrophils might be an important coeffector for the elimination of Mtb at the early stages of infection [61]. Moreover, an exceptional long-term vaccine effect was observed in the rhesus macaque model since the challenge of Erdman strain occurred as late as nearly one year after vaccination. Long-term vaccine efficacy was contributed by the lifelong persistent of the viral vector, and this unique characteristic makes CMV an optimal vaccine vector to deliver antigens.

## 5. CMV-Vectored Vaccine against Tumors

Cancer is a major threat to humans, the cause of high morbidity and mortality rates each year. Although exciting new progress has been made in recent years, humans are still powerless at dealing with most metastatic tumors. The development of prophylactic and therapeutic tumor vaccines has had a profound benefit to society, but the overall benefits derived from multiple candidate tumor vaccines used to prevent and treat cancer are limited. Tumor cells derived from normal cells with mutations activate oncogenes and/or inactivate tumor suppressor genes, which causes cell-cycle dysregulation and tumor formation. The human immune system is tolerant to self-antigens, while these mutations are too minor for the immune system to recognize and induce an effective immune response to eradicate tumor cells in most cases. Moreover, tumor-specific T cells tend to become anergic due to the immunosuppressive microenvironment in solid cancer. An optimal tumor-vaccine strategy should be to elicit effective antitumor immune responses that could eliminate tumor cells at the early stage, and be able to also evade the immunosuppressive mechanisms inside the tumor microenvironment [62].

Numerous laboratories and clinical studies have confirmed that CD8^+^ T-cell responses play important roles in the control of tumor progression. Tumor-infiltrative CD8^+^ T cells were proven to be positively related to the regression of solid tumors and overall survival improvement at all stages of the tumor [63,64,65]. Thus, the induction of a large number of effective antitumor T cells is an important indicator to evaluate the immunogenicity of a potential tumor vaccine. CMV can establish a latent infection with periodic reactivation, and repeated antigen stimulation elicits a large number of CMV-specific CD8^+^ T cells through “memory inflation” [11]. Natural CMV infection elicits broad T-cell responses specific to a wide range of antigens, and these HCMV-specific responses could comprise up to 10% of total memory CD4^+^ and CD8^+^ T cells [66]. This unique characteristic makes CMV an attractive vaccine vector to deliver tumor antigens. In a study, MCMV was used to express ovalbumin or modified melanoma antigen gp100, and these recombinant viruses were evaluated in a highly metastatic lung-melanoma model. Results showed both recombinant viruses could elicit robust and long-term tumor-antigen-specific polyfunctional CD8^+^ T-cell responses, and provide prophylactic and therapeutic protection against highly metastatic lung B16 melanoma [7]. In another study, investigators inserted NKG2D ligand *RAE-1γ* into the MCMV genome. This recombinant virus could be controlled by NK cells at the early stage of infection and was proved to be much safer, even for immunocompromised mice. The recombinant CMV virus could be used as a prophylactic or therapeutic tumor vaccine vector, and vaccination of this SIINFEKL epitope expressing RAE-1γ MCMV was able to induce SIINFEKL-specific CD8^+^ T cells and provide long-term protection to delay or prevent the growth of melanoma cells expressing ovalbumin [67]. A recent study has demonstrated that an MCMV-based vector encoding a single MHC-I restricted epitope on the C-terminal of the IE2 protein offered better immune protection than those encoding whole HPV E6 and E7 proteins in mice against the challenge with E6+E7-transformed TC-1 tumor cells [68]. Moreover, humoral immunity elicited by an MCMV-based vaccine vector expressing mouse tyrosinase-related protein 2 (TRP2) also contributed to prophylactic and therapeutic protection against melanoma [69].

The immunosuppressive microenvironment within solid cancers is another major barrier for the immune system to eliminate tumor cells. The mechanism of the tumor microenvironment is highly complex and it has not been clearly elucidated. The tumor microenvironment comprises multiple cell types, extracellular matrix and metabolic mediators [70]. This microenvironment forces the infiltrated tumor-specific T cells into an anergic state and makes them blind to tumor cells; thus, mitigation of the immunosuppressive microenvironment of solid tumors could cooperate with tumor-specific T cells in eliminating the tumor cells. Drugs or physical treatment, including chemotherapeutics and radiation, inhibitors of T-cell checkpoints, can mitigate the immunosuppressive microenvironment of solid tumors [71]. When working with a mouse melanoma model, researchers found that multiple melanoma-specific T cells induced by a CMV-based vaccine cannot effectively slow tumor growth, but direct injection of these viruses into the tumor could synergize with blockage of the PD-1 checkpoint to regress established tumors [72].

## 6. Evaluation of Immunogenicity of CMV-Based Vector in Animal Models

Understanding the immunology and pathogenesis of CMV is derived from both clinical studies and animal models. Direct in vivo studies on HCMV are fairly difficult because HCMV is strictly species-specific and can neither infect nonhuman primates and other animals, nor replicate in nonhuman cells. Moreover, CMV infection in healthy individuals is asymptomatic, and it is difficult to track the virus during the latent phase [73]. Thus, studies on species-specific CMV orthologs and related natural animal models are important to better understand HCMV biology and its associated immunology. MCMV, guinea-pig CMV, and RhCMV are natural animal pathogens most often used to investigate CMV-specific pathogenesis and immunity [74,75]. The overall viral life cycles and the function of many unique viral genes of these species-specific viruses overlap. Studies on these species-specific viruses do provide important translatable insights into HCMV pathogenesis and its role in immune-system activation.

Mouse and rhesus macaque models have been used to evaluate the immunogenicity of CMV-vectored vaccines in many studies. Results from mice vaccinated with MCMV/TB and rhesus macaques vaccinated with RhCMV/TB showed a vaccine-related innate response playing important roles in the control of Mtb infection. These results are coincident with some previous reports that innate immunity plays a role in trapping and killing Mtb because Mtb-antigen-specific T-cell responses are generally not thought to directly kill Mtb. However, compared with these nonhuman CMV species, some genes, especially genes encoding immunomodulatory HCMV proteins, demonstrate significant divergence during the long-term coevolution with humans; differences in host genetics might also affect the human translation of results derived from nonhuman CMV species [34]. Some important differences between the immune responses induced by RhCMV and HCMV were observed in the preliminary experiments. Unconventional CD8^+^ T-cell responses restricted by MHC-II and MHC-E were induced by the vaccination of fibroblast-adapted RhCMV68-1. These rare T-cell responses were believed to play an essential role in eliminating SIV in the early stage of infection since SIV could have not learned how to evade these unconventional immune responses. Although not well-understood, HCMV gH/gL/UL128–131 complex orthologs (Rh13.1, Rh61/Rh60, Rh157.5, Rh157.4 and Rh157.6 of RhCMV) were believed to play an important role in the induction of these unconventional CD8^+^ T cells because the pentamer-restored RhCMV (RhCMV68-1.2) was unable to induce these unconventional T-cell types and protect rhesus macaques from the SIV challenge [34,49]. In striking contrast, when a Towne/Toledo chimeric HCMV virus lacking the pentameric complex was tested in the clinical trial, this vaccine form was found to induce predominantly canonical MHC-I restricted CD8^+^ T-cell responses, and no MHC-II- or HLA-E (MHC-E homolog)-restricted T-cell responses were detected [76]. These results indicated that, during immunization of humans with an HCMV-based vaccine vector, unknown factors might also contribute to the induction of unconventional CD8^+^ T-cell responses.

Differences between the type of T-cell response elicited by RhCMV and HCMV indicate that an animal model that can be directly used to study the immunogenicity of HCMV is important for HCMV vaccine-vector engineering. Researchers implanted human fetal thymus/liver tissue to severe combined immunodeficient mice and constructed a xenograft, and this implanted human tissue could support the replication of HCMV in vivo. Low-passaged Toledo strain can establish infection in this implanted tissue, while fibroblast-adapted AD169 and Towne laboratory strain could not. These results were coincident with those of human challenge studies [10]. The xenograft mouse model is a useful tool to study viral pathogenesis, but it casts no light on the details of the elicited immune response [10,77]. An alternative animal model is the humanized bone marrow–liver–thymus mouse model. When these mice were injected with HCMV-infected fibroblast cells, HCMV-specific CD4^+^ and CD8^+^ T-cell responses, and HCMV-specific IgM and IgG neutralizing antibodies were detected [78]. Results indicated that a human adaptive immune system was reconstructed in the humanized mice. This mouse model is valuable to study HCMV-specific immune responses in the context of a functional human immune system. This mouse model could especially provide additional information for the T-cell type elicited by recombinant HCMV infection in the context of the discrepancy of CMV-elicited T-cell types in rhesus macaques and humans.

## 7. Safety Concerns of a CMV-Vectored Vaccine

Safety and immunogenicity are two-dimensional concerns for candidate vaccines. How to balance safety and immunogenicity is also important for designing a rational CMV vaccine vector. For a CMV-vectored vaccine, safety is the most important concern. Active CMV infection is associated with severe consequences in immunosuppressed patients that significantly increase mortality and morbidity rates [3]. Moreover, congenital CMV infections might lead to long-term neurodevelopmental sequelae, including mental retardation, seizure disorders, cerebral palsy, sensorineural hearing loss, microcephaly, and learning disabilities [4]. A vaccine vector should not come with an increased risk of congenital CMV infection or other serious diseases. It is also important to consider the immunogenicity of the viral vector when designing the CMV-based vector because the immunogenicity of the CMV-based vector also affects that of the protein it delivers. In general, an attenuated virus is required for a CMV vaccine vector. The Towne and AD169 laboratory strains are two well-characterized attenuated virus strains, and both were attenuated through extensive passages in fibroblast cells. Both fibroblast-adapted CMV viruses showed a fairly good safety profile in clinical trials [13,14]. However, the gH/gL/UL128-131 complex of this virus type is disrupted and the immunogenicity is compromised. The pentameric complex restored significantly increases the immunogenicity of the fibroblast-adapted virus, but retention of the pentameric complex in CMV might compromise the safety of the fibroblast-adapted virus. In a study, fibroblast-adapted guinea-pig CMV with a repaired pentamer-disrupting mutation restored its pathogenesis in immunosuppressed guinea pig, and this virus was able to cause a congenital infection [21]. In the design of Merck’s vaccine, ddFKBP was introduced to conditionally degrade two essential viral genes and rendered the virus conditionally replication-defective [23]. This strategy balances the safety and immunogenicity of the vaccine well, but further considerations are required when using CMV as a vaccine vector. Life-long low-level persistence of CMV makes it a rational, optimal, and unique vector to deliver antigens, and the strategy might elicit prolonged and self-boosted immunity due to asymptomatic reactivation. Prolonged immunity enables the immune system to clear the evaded pathogens at the early stage of infection, and this characteristic is important for the elimination of pathogens that have evolved immune-evasion strategies. This conditionally replication-defective virus cannot generate a progeny virus and only has one life cycle in vivo. The virus could not establish latency and reactivate after vaccination, and the ability to elicit prolonged immunity might be compromised. One solution to this problem might rely on further understanding the CMV gene function. CMV has multiple immunomodulatory genes, and the function of most of these genes has not been fully elucidated. A better understanding of these genes might provide new strategies to create a novel vaccine vector. An alternative solution is to set a rational immunization plan. This strategy relies on follow-up data from the Merck CMV vaccine. A boost at the appropriate time could enhance waning immunity and elongate the protection that the vaccine provided.

Another safety concern is related to the potential tumorigenic ability of CMV. CMV is a lytic virus to most permissive cells and it is generally not regarded as a tumor virus. When primary infection happens in a healthy individual, systematic dissemination might happen, and the virus eventually establishes latency in hematopoietic progenitor cells or monocyte/macrophages; the presence of the virus in other types of tissue is under the detection limit [79]. However, although controversial, viral DNA, mRNA, and antigens have been detected in different types of tumor tissue in several recent reports, suggesting that CMV infection plays a role in the etiology of these tumors [80,81,82]. Accumulated data showed that many HCMV proteins could activate multiple pro-oncogenic pathways and play a role in evading tumor suppressors, enhancing cellular proliferation and immortality, increasing host-cell genome instability and cell survival, and deregulating cellular energetics [81,83]. As with well-known human tumor viruses, CMV proteins could interfere in the function of tumor-suppressor proteins p53 and pRb, and drive the cell cycle to pass the Restriction Point (R point). Suppression results in the dysregulation of cell division and apoptosis [83]. Due to the distinct behavior of CMV in normal permissive cells and tumor cells, the concept of oncomodulation was proposed to explain the role of CMV found in tumor tissue. Instead of lysing permissive cells, CMV infection could increase the malignancy of tumor cells. Several CMV proteins were found to be involved in the process, for example, viral product cmvIL-10 was found to be able to bind to the cellular IL-10 receptor and activate *STAT3*. Constitutive activation of *STAT3* might increase the metastasis and chemoresistance of tumor cells in malignant cancer [84,85,86]. To ease the safety concerns of potential tumorigenicity, engineering a CMV-based vector might be required to eliminate some metastasis- and chemoresistance-related genes. Evaluation of a CMV-based vector in tumor patients might also be necessary when testing in clinical trials.

## 8. Discussion

The first time of proposing CMV as a vaccine vector to encode fertility-associated proteins such as zona pellucida or sperm antigens in order to induce immunocontraceptive can be traced back to as early as 1994. MCMV encoding the zona pellucida antigen was eventually proven to be able to sterilize female mice in a study performed nine years later [87,88]. CMV vaccine vector attracted much attention until a work performed in 2011, in which investigators proved a rhesus CMV-encoding SIV antigens can profoundly protect rhesus macaques against a highly pathogenic SIV challenge [44]. CMV vectors were found to be able to induce conventional “non-canonical” MHC-I-restricted and unconventional MHC-II- and MHC-E-restricted T cells recognizing unusual, diverse, and highly promiscuous epitopes [45,49]. The discovery of special CD8^+^ T-cell responses induced by CMV-based vectors shed light to novel prophylactic and therapeutic vaccine development, especially to pathogens that are capable of escaping host natural immunity and establishing repeated infection or chronic infection [48]. The efficacy of the CMV-based vaccine vector was further tested against different tumors, malaria, Mtb infection, Ebola virus, and respiratory-syncytial-virus (RSV) infection in various animal models [7,60,69,89,90,91,92]. All these works showed promising efficacy when using CMV as a vaccine vector.

Comparing with other viral vaccine vectors (such as Ad, vaccinia virus, and MVA), CMV has some unique and promising characteristics. Viral vectors are able to efficiently deliver foreign genes in vivo and elicit robust immunoresponses, while a CMV-based vector has the additional advantage of eliciting at least four distinct types of CD8^+^ T-cell responses through genetic engineering [55]. These highly diverse CD8^+^ T-cell responses were contributed by CMV components through known or unknown mechanisms [55]. Unconventional T-cell responses were not seen in Ad or MVA vectors expressing the same antigen. CD8^+^ T-cell responses elicited by Ad or MVA vector expressing SIV proteins are similar with SIV infection; conventional T-cell responses (canonical MHC-I restricted) failed to provide protection for rhesus macaques since SIV had evolved strategies to evade these immunoresponses [46,49]. Unconventional CD8^+^ T-cell responses elicited by RhCMV/SIV were believed to protect rhesus macaques against the challenge of highly pathogenic SIVmac239 virus [55]. Another unique characteristic of the CMV-based vector is that CMV can establish latency after primary infection and reactivate asymptomatically in healthy individuals. Reactivated viruses are able to boost immunity and maintain a high frequency of non-exhausted effector T cells in circulation. These immunoresponses would continuously monitor the entry and early dissemination of pathogens, and intercept infection at the early stage [55]. Moreover, these effector memory T cells facilitate the elimination of tumors, parasites (plasmodium), and bacteria (Mtb) [7,60,91].

Pre-existing immunity, specific to the vector itself, is a major limitation to hamper the delivery of foreign genes through a viral vaccine vector. Vector-specific immunity reduces the transduction efficiency of a viral vaccine vector and limits the duration of the target gene expression. Conventional viral vectors (such as Ad and vaccinia virus) are usually eliminated by pre-existing immunity before providing meaningful efficacy [93,94]. However, results from multiple studies supported that the efficiency of a CMV-based vector should not be fluctuated by pre-existing immunity: sequential infection of immunocompetent mice with multiple strains of murine CMV could cause a mixed infection; RhCMV vectors expressing SIV antigens could persistently infect rhesus macaques and elicit robust SIV-specific T-cell responses regardless of pre-existing RhCMV immunity; secondary infection or reactivation of CMV could occur in women with preconceptional immunity [12,46,95].

However, exploration of CMV as a novel vaccine vector is still at an early stage, and there are many questions still unanswered. Although CMV-based vaccine vectors could elicit protective immune responses against a variety of viral, bacterial, parasitical, and tumor targets, protective determinants seem to vary in these challenges. Fibroblast-adapted RhCMV68-1 was reported to induce at least four different types of CD8^+^ T cells, and these rare T-cell responses were believed to be important for protecting rhesus macaques against a highly pathogenic SIV challenge, while gH/gL/UL128-131 complex-restored RhCMV68-1.2 mainly induced conventional canonical MHC-I-restricted CD8^+^ T cells and failed to afford protection [45,49]. It suggests that an induction of unconventional CD8^+^ T-cell response is very important for the prevention of HIV infection. In striking contrast, both the RhCMV68-1- and the RhCMV68-1.2-based vectors afforded similar protection for rhesus macaques against the Mtb challenge, and this protection was independent of unconventional CD8^+^ T-cell responses. Although not well elucidated, both CD8^+^ T cells and innate immunity, especially NK-cell or neutrophil activation, were believed to contribute to the protection [61]. In two separate studies, MCMV was used to encode the zona pellucida antigen or TRP2. The former recombinant virus was able to sterilize female mice, while the latter induced the rejection of melanoma in mice; the efficacy of both vaccine types was demonstrated to be dependent on antigen-specific antibody responses [69,96]. In other challenge studies of most tumors, malaria, RSV, and the Ebola virus, protections were dependent on CD8^+^ T-cell responses, elicited by CMV-based vaccine vector [55,58,81,82,83,84].

A CMV-based vaccine vector might be a promising vector for the protection and cure of more prominent diseases such as Alzheimer’s disease and glioblastoma [97]. To ensure CMV could be used as a safe and immunogenic vaccine vector, further studies on animal models and in clinical trials are required to better understand the immunity elicited by the CMV-based vaccine vector. Moreover, understanding the function of viral genes and the role of genes taking part in inducing different sets of T-cell responses would also provide useful information for choosing CMV virus strains and making further modifications.

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
