# Peer review of "Promising Cytomegalovirus-Based Vaccine Vector Induces Robust CD8+ T-Cell Response"

_ijms, 2019, doi:10.3390/ijms20184457_

Round 1

Reviewer 1 Report

The manuscript will need checking for grammar - i have highlighted a few areas below where clarification is required

Check the spellings of non-attenuated' overattenuated etc. Sometimes a space is used and sometimes not between the qualifier (non; un or over) and the word attenuated. Edit for consistency.

Ln151 the determinant of attenuation - this is a big claim may be needs to be better phrased or better explained

Ln 153 why not just pentamer rather than gH/gL pentamer?

Ln 440 - 'kept trying' change to 'attempted'

Ln 442 - Define ENV (is it not Env?)

Ln 647 - remove 'a' and add virus after SIV39mac

Ln 650 post challenge (space)

Ln652-653 dont follow the argument why was the detection of dissemination post challenge evidence of clearance of the viral reservoir?

Ln675 have Ad and MVA been previously defined

Author Response

Responses to Reviewers’ Comments for manuscript IJMS-584359

On behalf of my co-authors, we thank you very much for giving us an opportunity to revise our manuscript.

Responses to reviewer #1:

The manuscript will need checking for grammar - i have highlighted a few areas below where clarification is required

Check the spellings of non-attenuated' overattenuated etc. Sometimes a space is used and sometimes not between the qualifier (non; un or over) and the word attenuated. Edit for consistency.

Response: Thank you for your constructive comment. We revised these and other similar expressions, please refer to line 42, 49, 67, 74, 78, 86, 233, 255, 259, 456.

Ln151 the determinant of attenuation - this is a big claim may be needs to be better phrased or better explained

Response: Thank you for your comment. We changed the expression with “a significant contributor to attenuation”. The expression was used in the paper we cited (Mcvoy, et al. 2016). Please refer to Line 90, 93-94.

Ln 153 why not just pentamer rather than gH/gL pentamer?

Response: Thank you for your comment. We revised the expression. Please refer to Line 97.

Ln 440 - 'kept trying' change to 'attempted'

Response: Thank you for your comment. We revised it. Please refer to Line 146

Ln 442 - Define ENV (is it not Env?)

Response: Thank you for your comment. Yes, it refered to Env. We revised it. Please refer to Line 148-149.

Ln 647 - remove 'a' and add virus after SIV39mac

Response: Thank you for your comment. We revised the expression and also added virus after SIVmac239 at several other places. Please refer to Line 168, 169, 258, 478.

Ln 650 post challenge (space)

Response: Thank you for your comment. We revised it. Please refer to Line 171.

Ln652-653 dont follow the argument why was the detection of dissemination post challenge evidence of clearance of the viral reservoir?

Response: Thank you for your comment. Yes, we understand clearance of the lentiviral reservoirs is a milestone of HIV treatment, and almost all current therapy including ART cannot clear the viral reservoir. Any claims of clearance of the lentiviral reservoirs should provide extensive and definite evidences. In the study (Hansen, et al. 2013), the author listed quite a lot of evidences, which were not fully listed in our reviews. The author observed lymphatic and haematogenous viral dissemination after challenge with SIVmac239 virus, and replication-competent SIV persists in several sites for weeks to months. These results might indicate SIVmac239 had established systematic infection and latent infection. Rhesus got protected lost signs of SIV infection over time, and no measurable plasma- or tissue-associated virus could be detected using ultrasensitive assays; No SIV RNA or DNA sequences could be detected from tissues of protected rhesus necropsied 69-172 using ultrasensitive quantitative PCR method; No replication-competent SIV could be detected by extensive co-culture analysis of tissues or by adoptive transfer of 60 million haematolymphoid cells to naïve rhesus. All these evidences showed at least some lentiviral reservoirs might be susceptible to the immune responses elicited by RhCMV/SIV.

However, It might need more repeated experiments and more group of experiments to support the claim of clearance of the viral reservoir. The results are interesting and important, We hope to see the result of follow-up experiments. Anyway, we accepted your comment and revised the description. Please refer to Line 172-174.

Ln675 have Ad and MVA been previously defined

Response: Thank you for your comment. We had already defined Ad and MVA. Please refer to Line 189-190.

Reviewer 2 Report

In the manuscript entitled ‘Promising Cytomegalovirus-Based Vaccine Vector Induces Robust CD8+ T-Cell Response’ the authors summarize and discuss the implications of publications related to the development of CMV-targeted vaccine as well as CMV-vectored vaccines. All in all the authors do a thorough job discussing the data presented from work with HCMV, RhCMV, MCMV and GPCMV and cover the different types of vaccine strategies and types of immune responses elicited. Some relatively minor comments are listed below.

A better distinction between the needs of a vaccine against CMV and the use of CMV as a vaccine vector should be presented in the introduction. It is jarring to read about the work towards a vaccine against CMV, due to its pathology during congenital infection and in immunosuppressed patients, and in the next paragraph how it is a promising vaccine vector. Section 2 and 3 should begin with a brief recap of these distinctions as well. Line 132 refers to only 59% of patients receiving ART – what patient population is this referring to? Patients worldwide or in a specific country/region? Within the section referring to CMV vaccination against SIV, the manuscript may benefit from a clear description of what is being referred to as conventional versus unconventional and canonical versus non-canonical T cell responses. Later in the manuscript this is clearly stated but should be present here when the concept is first introduced. The paragraph from lines 191-201 discuss the role of US6 family members in blocking conventional CD8+ T cell responses. The authors erroneously state: Unconventional MHC-II-restricted and conventional MHCI-restricted noncanonical CD8+ T cell-responses are rarely induced by other viral-vector platforms due to the special role of the CMV US6 gene-family homologs. CMV US6 does not play a role in the elicitation of MHC-II-restricted or MHC-E-restricted CD8+ T cells (these responses are still elicited in the absence of US6 family members). Its role is to specifically block canonical MHC-Ia-restricted CD8+ T cells. The discussion of the AD169 antibody induction on lines 205-209 seems misplaced. In general, there are several instances of repetitiveness, where a concept that was already introduced earlier in the manuscript is restated again. The manuscript would benefit with more careful editing. As well, there are some minor grammatical errors throughout the manuscript.

Author Response

On behalf of my co-authors, we thank you very much for giving us an opportunity to revise our manuscript.

Responses to reviewer #2:

In the manuscript entitled ‘Promising Cytomegalovirus-Based Vaccine Vector Induces Robust CD8+ T-Cell Response’ the authors summarize and discuss the implications of publications related to the development of CMV-targeted vaccine as well as CMV-vectored vaccines. All in all the authors do a thorough job discussing the data presented from work with HCMV, RhCMV, MCMV and GPCMV and cover the different types of vaccine strategies and types of immune responses elicited. Some relatively minor comments are listed below.

A better distinction between the needs of a vaccine against CMV and the use of CMV as a vaccine vector should be presented in the introduction. It is jarring to read about the work towards a vaccine against CMV, due to its pathology during congenital infection and in immunosuppressed patients, and in the next paragraph how it is a promising vaccine vector.

Response: Thank you for your comment. The serious consequences caused by CMV infection limit to some specific people and at some specific states, CMV infection is asymptomatic in most healthy people. When using CMV as a vaccine vector, the potential risk caused by CMV infection is able to control. CMV virus is easy to be attenuaed and the safety can be improved through viral genome engineering. Fibroblast-adapted Towne and AD169 strain have been tested in numerous clinical trials and showed excellent safety. Towne and AD169-based or other rational designed vaccine vector should be fairly safe. We added a sentence before the introduction of CMV vaccine vector. Please refer to Line 44-46.

 Section 2 and 3 should begin with a brief recap of these distinctions as well.

Response: Thank you for your comment. We added brief introduction at the beginning of Section 2 and 3. Please refer to Line 61-63 and Line 162-163.

Line 132 refers to only 59% of patients receiving ART – what patient population is this referring to? Patients worldwide or in a specific country/region?

Response: Thank you for your comment. The data was calculated from patients worldwide and it came from WHO. We revised the sentence. Please refer to Line 140.

Within the section referring to CMV vaccination against SIV, the manuscript may benefit from a clear description of what is being referred to as conventional versus unconventional and canonical versus non-canonical T cell responses. Later in the manuscript this is clearly stated but should be present here when the concept is first introduced.

Response: Thank you for your comment. We defined these concepts as your suggestions. Please refer to Line 194-199.

The paragraph from lines 191-201 discuss the role of US6 family members in blocking conventional CD8+ T cell responses. The authors erroneously state: Unconventional MHC-II-restricted and conventional MHCI-restricted noncanonical CD8+ T cell-responses are rarely induced by other viral-vector platforms due to the special role of the CMV US6 gene-family homologs. CMV US6 does not play a role in the elicitation of MHC-II-restricted or MHC-E-restricted CD8+ T cells (these responses are still elicited in the absence of US6 family members). Its role is to specifically block canonical MHC-Ia-restricted CD8+ T cells.

Response: Thank you for your comment. We checked it carefully and you are right, US6 gene-family homologs do not play roles in regulating these unconventional CD8+ T cell responses. We revised the sentence. Please refer to Line 215. Thank you very much!

The discussion of the AD169 antibody induction on lines 205-209 seems misplaced.

Response: Thank you for your comment. Actually, we want to describe two distinct roles the pentamer plays in the vaccine efficacy. In the researches of HCMV, most researches revealed the role of the pentamer plays in eliciting robust neutralizing antibodies. While in the case of RhCMV, the pentamer was also found to affect the types of T-cell response. We revised the description for better understanding. Please refer to Line 224-228.

In general, there are several instances of repetitiveness, where a concept that was already introduced earlier in the manuscript is restated again. The manuscript would benefit with more careful editing. As well, there are some minor grammatical errors throughout the manuscript

Response: Thank you for your comment. We deleted some repeated contents. Please refer to Line 468-469, 476, 500-502.

We revised the grammatical errors raised by Reviewer 1. Please refer to “Responses to reviewer #1”.

We also found and revised some errors during editing. Please refer to Line 25, 57, 73, 85-86, 174-176, 201, 272.
